# VEGF and Other Gene Therapies Improve Flap Survival—A Systematic Review and Meta-Analysis of Preclinical Studies

**DOI:** 10.3390/ijms25052622

**Published:** 2024-02-23

**Authors:** Wiktor Paskal, Mateusz Gotowiec, Albert Stachura, Michał Kopka, Paweł Włodarski

**Affiliations:** 1Department of Methodology, Medical University of Warsaw, 1b Banacha Street, 02-091 Warsaw, Poland; mateusz.gotowiec@wum.edu.pl (M.G.); michal.kopka@wum.edu.pl (M.K.); pawel.wlodarski@wum.edu.pl (P.W.); 2Doctoral School, Medical University of Warsaw, 81 Żwirki i Wigury Street, 02-091 Warsaw, Poland

**Keywords:** surgical flaps, regenerative medicine, animal models, gene therapy, genetic vectors

## Abstract

Surgical flaps are basic tools in reconstructive surgery. Their use may be limited by ischemia and necrosis. Few therapies address or prevent them. Genetic therapy could improve flap outcomes, but primary studies in this field present conflicting results. This systematic review and meta-analysis aimed to appraise the efficacy of external gene delivery to the flap for its survival in preclinical models. This review was registered with PROSPERO (CRD42022359982). PubMed, Embase, Web of Science, and Scopus were searched to identify studies using animal models reporting flap survival outcomes following any genetic modifications. Random-effects meta-analysis was used to calculate mean differences in flap survival with accompanying 95% CI. The risk of bias was assessed using the SYRCLE tool. Subgroup and sensitivity analyses were performed to ascertain the robustness of primary analyses, and the evidence was assessed using the GRADE approach. The initial search yielded 690 articles; 51 were eventually included, 36 of which with 1576 rats were meta-analyzed. VEGF gene delivery to different flap types significantly improved flap survival area by 15.66% (95% CI 11.80–19.52). Other interventions had smaller or less precise effects: PDGF—13.44% (95% CI 3.53–23.35); VEGF + FGF—8.64% (95% CI 6.94–10.34); HGF—5.61% (95% CI 0.43–10.78); FGF 3.84% (95% CI 1.13–6.55). Despite considerable heterogeneity, moderate risk of bias, and low quality of evidence, the efficacy of VEGF gene therapy remained significant in all sensitivity analyses. Preclinical data indicate that gene therapy is effective for increasing flap survival, but further animal studies are required for successful clinical translation.

## 1. Introduction

Surgical flaps comprise most of the alternatives to the reconstructive ladder. They offer complex tissue defect coverage along with potential muscular, vascular, and neural function donors to the recipient area. Thus, they are often superior to other reconstructive options. Flap harvesting requires sacrificing a tissue bulk from the donor area, and flap survival relies on extensive vascular support in the recipient region. As a result, flaps are prone to postoperative complications (ischemia, venous congestion, and lymphatic drainage dysfunction). Necrosis usually begins distally to the flap pedicle. This is related to insufficient angiogenesis, which depends on cytokines and nutrients delivered from the pedicle. As a result, the distal parts are not only vulnerable to nutrient deprivation and ischemia but also to ischemia-reperfusion injury, which is characterized by the influx of free radicals to nutrient-poor regions shortly after establishing blood flow. Depending on the flap type used, in the clinical settings, 5–20% ischemia or necrosis rate is reported, while preclinical models produce necrosis in 20–80% of the flap area.

Numerous attempts have been made to decrease flap necrosis, particularly in patients with vascular comorbidities who are at a higher risk of revisions and flap loss [1]. Since most flap-based reconstructions are planned surgeries, various preconditioning interventions have been implemented to improve survival. However, effective techniques are limited to surgical delay, local hyperthermia or hypothermia treatment, and pharmacological preconditioning [2]. Intraoperative techniques include the use of vessel supercharging techniques or intraoperative preconditioning. Postoperative ischemia treatment may be addressed by hyperbaric oxygen treatment, local vasodilator application, or topical growth factor treatment [3,4]. Although such interventions can provide short-term increases in vessel density in flap surroundings, they require repetitive administration (except surgical delay) and strongly depend on accessibility to the affected flap tissue. Finally, these techniques do not provide localized support to the distal parts of the flap but rather affect the whole tissue bulk, which can further increase the risk of ischemia-reperfusion injury. Most of them focus on indirect stimulation of angiogenesis. Apart from the surgical delay and hypo/hyperthermia therapy of the panned flap, no other therapies could be implemented before or during flap elevation. Finally, most of the current therapies aim to decrease the area of yet-present necrosis or ischemia.

External delivery of proangiogenic particles is complicated, and repetitive administration methods may increase complication rates. To overcome these limitations, genetic engineering tools have emerged. They allow more localized and specific biological drug delivery, together with long-term target protein production within the tissue [5]. The first report of improved hind limb survival via VEGF-plasmid transfection in a rabbit model was presented by Takeshita in 1996 [6]. Thereafter, a wide variety of target proteins, transformation techniques, and applications in surgical flap animal models have been explored. 

To date, there has been no systematic summary of the efficacy of surgical flaps’ genetic modification before, during, or after their elevation. To address this knowledge gap, we performed a systematic review and meta-analysis to assess flap survival outcomes following these interventions in animal models.

## 2. Results

The initial literature search yielded 690 papers, reduced to 644 after deduplication. After abstract screening, 76 articles were eligible for full-text retrieval. Eventually, 51 studies [7,8,9,10,11,12,13,14,15,16,17,18,19,20,21,22,23,24,25,26,27,28,29,30,31,32,33,34,35,36,37,38,39,40,41,42,43,44,45,46,47,48,49,50,51,52,53,54,55,56,57] were included in the qualitative synthesis (Table 1), of which 36 studies with 1576 rats were analyzed quantitatively (Figure 1).

### 2.1. Primary Outcome—Flap Survival

#### 2.1.1. Vascular Endothelial Growth Factor (VEGF)

The authors used VEGF as an intervention in 27 studies and 45 comparisons (Figure 2) [7,8,9,10,11,12,13,15,16,28,29,30,32,33,34,37,38,39,40,41,42,43,45,46,47,48,50]. Eight studies used viruses as vectors, five used cells, and 13 used plasmids. The overall effect estimate for this intervention was a mean increase in flap survival area of 15.66% (95% CI 11.80–19.52), and the results were highly heterogeneous (I^2^ = 96%). 

There were no subgroup differences between the vectors used (χ^2^ = 2.55, df = 2, *p* = 0.28), with all subgroups presenting similar heterogeneity. A subgroup analysis based on flap type showed that two thirds of the comparisons studied were performed using McFarlane or epigastric flaps. We detected significant subgroup differences with the highest effect estimate in comparisons using TRAM flap (MD 19.68, 95% CI 10.70–28.65, k = 4 studies, I^2^ = 56%) and the lowest in musculocutaneous flap (MD 7.41, 95% CI −0.15–14.96, k = 3 comparisons (1 study, I^2^ = 95%). The overall effect estimate did not change significantly after sensitivity analyses (Appendix A). We performed a subgroup analysis based on the delivery route (Appendix A) and found significant subgroup differences (*p* < 0.001). To identify potential effect moderators, we used meta-regression considering administration time, assessment time, vector type, and administration route (Appendix A); however, the test of moderators was insignificant (*p* = 0.39), and our model accounted for only 4.27% of total heterogeneity. We did not identify significant publication bias in this group of studies (Appendix A).

#### 2.1.2. Fibroblast Growth Factor (FGF)

This intervention was used in 6 studies and 12 comparisons (Figure 3) [13,17,23,31,39,49]. One study used viruses as vectors, three used plasmids, and two used cells. Interventions with FGF, on average, increased the flap survival by 3.84% (95% CI 1.13–6.55), and there was considerable heterogeneity between studies (I^2^ = 87%). This effect diminished and became insignificant after sensitivity analyses. 

Only the subgroup with a cellular delivery system showed a significant improvement in flap survival (MD 5.99, 95% CI 2.51–9.47). In the two remaining subgroups, improvement was small and insignificant. All studies utilizing a cellular vector had a high risk of bias. Overall, most studies used McFarlane flaps. Flaps’ survival in this group was improved by FGF administration by a mean of 4.65% (95% CI 1.50–7.79)–which was markedly diminished and rendered insignificant after performing sensitivity analyses (Appendix A). For TRAM and dorsal island skin flaps, FGF did not improve the outcome compared to placebo. We did not identify a significant publication bias in this group of studies (Appendix A).

#### 2.1.3. Platelet-Derived Growth Factor (PDGF)

Platelet-derived growth factor was used in 4 studies and 5 comparisons (Figure 4) [11,17,18,44]. Three studies used plasmids as vectors and one used cells. PDGF treatment increased the flap survival by an average of 13.44% (95% CI 3.53–23.35) with an overall considerable heterogeneity (I^2^ = 92%). This effect became marginally insignificant (MD 10.94, 95% CI −0.35–22.24) after performing sensitivity analyses (Appendix A). We only found a significant effect in a subgroup of studies using plasmids (MD 13.75, 95% CI 4.31–23.19) and those utilizing McFarlane flaps (MD 8.78, 95% CI 4.72–12.84). Both effects remained stable and significant after performing the sensitivity analyses (Appendix A).

#### 2.1.4. Hepatocyte Growth Factor (HGF) and HGF + Prostacyclin Synthase (PGIS)

Hepatocyte growth factor alone or enriched with prostacyclin synthase was used in three studies and four comparisons, and one study and four comparisons, respectively (Figure 5) [19,20,53]. Sole HGF treatment increased flap survival by an average of 5.61% (95% CI 0.43–10.78) with moderate heterogeneity (I^2^ = 52%). An addition of PGIS insignificantly increased flap survival to an average of 8.16% (95% CI −1.09–17.41). Overall, any therapy containing HGF increased flap survival by an average of 6.53% (95% CI 1.59–11.48) with a substantial heterogeneity of included studies (I^2^ = 64%).

#### 2.1.5. VEGF + FGF

Combined therapy with VEGF and FGF was used in 2 studies and 5 comparisons (Figure 6) [13,49]. On average, it increased flap survival by 8.64% (95% CI 6.94–10.34) with considerable heterogeneity (I^2^ = 80%).

#### 2.1.6. Other Interventions

Apart from the studies included in the quantitative analysis, we found several studies with various treatment types used in less than three comparisons. Other substances or combinations containing unidentified molecules were used in 15 studies [14,15,21,24,25,26,27,35,36,51,52,54,55,56,57]. The substances or targets previously described include hypoxia-inducible factor, Dikkopf-2, nitric oxide synthase, interleukin-10, angiopoietin, relaxin, NF-κB, KGF, and SDF-1. Studies used a variety of different vectors: plasmids (four studies), viruses (seven studies), cells (three studies), and oligonucleotides (one study). Overall, all types of interventions increased flap survival, with treatment using anti-NF-κB oligonucleotide achieving the best results and treatment with KGF-containing virus, the lowest.

### 2.2. Secondary Outcome–Vessel Density

The authors measured vessel density using a variety of methods in 13 studies and 35 comparisons (Appendix A) [7,11,12,13,15,20,33,41,42,45,47,49,53]. The overall effect estimate for all intervention types was an increase in standardized mean difference of 5.25 (95% CI 3.62–6.87), results were considerably heterogeneous (I^2^ = 93%), and intervention subgroups differed (*p* < 0.01) (Figure 7).

We performed a meta-regression considering administration time, assessment time, and type of intervention (Appendix A) with a significant test of moderators (*p* = 0.03). Assessment time (SMD 1.0, 95% CI 0.38–1.63) and VEGF intervention (SMD 4.85, 95% CI 0.08–9.62) were significant positive moderators of the effect size. The model accounted for 9.10% of total heterogeneity. There was a significant publication bias reflected in a funnel plot asymmetry and a significant (*p* < 0.001) Egger’s test result.

### 2.3. Risk of Bias

Studies were assessed using the SYRCLE Risk of Bias Tool for Animal Studies. The average score was 4.34 out of 10 points (Appendix A). The most frequently reported sources of potential bias included the lack of random housing of animals (*n* = 49), blinding following intervention (*n* = 49), outcome assessor blinding (*n* = 45), allocation concealment (*n* = 42), allocation sequence randomization (*n* = 37), and random animal selection for outcome measurements (*n* = 39). Few studies did not address incomplete outcome data (*n* = 7). Other sources of bias were also found in some of the studies (*n* = 11). This included imprecise data presentation, a low number of animals/flaps in the experimental groups, dubious lack of animal loss, uncommonly low SD values, and no clear definition of primary and secondary outcomes. 

### 2.4. Certainty of Evidence 

The assessment of certainty of evidence using GRADE has shown that only VEGF treatment may improve flap survival and vessel density (Appendix A). For all other interventions, it is uncertain if they improve any outcomes as the certainty of evidence is very low.

## 3. Discussion

### 3.1. Main Findings 

We found that treatment with locally administered growth factors delivered via vectors increased surgical flap survival in vivo by between 4% (FGF) and 16% (VEGF). The primary outcome was most assessed on day 7. The studied interventions also enhanced angiogenesis, increasing vessel density by an SMD of over 5, a time-dependent effect most prominently observed with VEGF. Although many studies were included in this meta-analysis, identifying the key factors influencing the outcome effect estimates is challenging. The levels of statistical heterogeneity were substantial or considerable for most analyses and could arise from differences in study design, systematic error (bias), applied interventions, and random error. 

Subgroup analysis showed that VEGF administration had the highest effect estimate for flap survival improvement in muscular flaps (TRAM) among other flap types. However, this effect was the least precise. Thus, we can only hypothesize that the underlying mechanism may be related to poor ischemia tolerance in muscular flaps, resulting in greater improvements if treated [58]. 

Seemingly important study characteristics, such as the route of administration, the timing of the intervention, or the vectors used, failed to explain the differences in effect sizes between studies, as the meta-regression showed. Only 9% of total heterogeneity among studies using VEGF could be accounted for, taking the above-mentioned factors into consideration. Different surgical flaps were used across studies; however, in most cases, they showed unanimously beneficial effects of locally overproduced growth factors.

Despite inconclusive results from meta-regression concerning effect moderators, several studies have provided indicative data.

Delivery time is an important determinant for the further translation of the results. We lack high-quality studies with head-to-head comparisons of different preoperative administration times, as only two studies of acceptable quality addressed this issue [30,34]. Giunta et al. [30] indicated that increasing the delay between adenoviral administration and flap elevation by up to 7 days is beneficial. Chen et al. [34] used modified myoblasts and found that using intervention on preoperative day 4 resulted in better outcomes than on preoperative days 0, 2, or 7. They relied on same-animal internal controls, unlike most of the studies, which may explain the relatively narrow confidence intervals. As for the general characteristics of viral transduction [59], if viral or cellular delivery is planned, some delay in flap harvesting should be considered (4–7 days). 

We lacked data on the optimal flap harvesting time following plasmid-based gene therapy. Based on the summarized results, we hypothesized that if the aim of the therapy was to prevent flap ischemia in the early postoperative period, the intraoperative use of a plasmid containing VEGF may be considered. However, further deductions are limited, as details concerning promoter and plasmid delivery form (e.g., liposomes, electroporation, etc.) were not considered in the meta-analysis. In both random and free flaps, direct tissue injections (subcutaneous, intradermal, and intramuscular) were used most frequently. Free flap or pedicled flap use enables consideration of intravascular administration of genetic therapy. Only two studies [38,51] used intra-arterial perfusion of flaps to administer either VEGF or anti-NF-κB oligonucleotides, both achieving high flap survival. However, single studies with a high risk of bias, using different interventions, are insufficient to confirm the superiority of this administration route. In addition, the advantages of intravascular therapy should be considered alongside potential risks (yet unreported) of thrombosis after injury to the pedicle, irritation of the intima with a high concentration of particles, and, by default, limited use of certain flap types.

The isoforms of target proteins should not be overlooked. Rinsch et al. [39] showed that VEGF-167 is significantly more efficient than the other forms of VEGF. This may be related to its ischemia-specific angiogenic activity previously observed in the myocardium and its role as a fatty acid metabolism regulator [60]. Compared with VEGF-165, which is the main angiogenesis regulator, VEGF-167 may be more specific to unstable conditions in the flap microenvironment.

Using multiple proteins does not seem to improve flap survival, as combined VEGF + FGF treatment was not superior to other interventions. This finding is limited by the small number of included studies. Although these cytokines are known to work synergistically [61], the lack of a clinically meaningful synergistic effect may be explained by the delivery mode resulting in metabolic overactivation following intense plasmid transfection.

Meta-regressions and sensitivity analyses did not prove that any of the parameters related to vectors—either cells, viruses, or plasmid produce a significant difference for the primary outcome. Thus, further research on pharmacodynamics and pharmacokinetics should be designed. 

Interestingly, only Uemura et al. [51] reported the use of intra-arterially administered oligonucleotides, showing a gap in knowledge regarding this treatment modality. As this treatment provided the best results among the qualitatively described studies, further investigation based on this vehicle and the route of delivery is warranted [62].

Currently, there are 32 FDA-approved cellular or gene therapies, proving that contemporary techniques are sufficiently safe and efficient for clinical use [63]. The greatest controversies and concerns related to gene therapy are related to potential systemic cyto- and genotoxicity or unpredictable distant side effects in nontarget locations. However, none of the included studies reported such.

### 3.2. Limitations of the Study

Although a beneficial effect was clearly observed, the results showed substantial statistical heterogeneity and at least a considerable risk of bias. Such heterogeneity is not uncommon in preclinical studies and can be explained by the lack of unified housing conditions and the varying strains of animals used. It is noteworthy that studies were conducted over a relatively wide period of over 20 years, during which standards of conducting and reporting animal studies have altered significantly, especially in the European Union, due to novel regulations [64]. Studies included in this review were generally of poor quality as per the risk of bias assessment, mostly due to inappropriate randomization and blinding. Relatively low SDs may require further consideration of the quality and reliability of reported data. This phenomenon was explicitly observed in studies with a low animal/flap count. A high risk of bias in the selected studies, coupled with surprisingly narrow confidence intervals, is likely the main reason for the considerable statistical heterogeneity. They would also explain why the meta-regression model accounted for a small percentage of the total heterogeneity. As this was the first meta-analysis in the field, we did not want to restrict the inclusion criteria. Access to comprehensive data reports was limited, and most studies did not publish supplementary datasets. The results had to be extracted based on the values found in the main text or directly from the figures. 

Overall, despite the beneficial effects of some molecules (particularly VEGF), the quality of evidence is low to very low, undermined mainly by a high risk of bias in the included studies and differing levels of random error.

## 4. Methods

This study was undertaken in accordance with the preferred reporting items for systematic reviews and meta-analyses (PRISMA) guidelines [7], and the study protocol was registered in the PROSPERO database (CRD42022S359982). PRISMA checklist is available in Appendix A.

### 4.1. Search Strategy

A comprehensive and systematic search of PubMed, Embase, Web of Science, and Scopus was conducted on 11 December 2022 and updated on 8 January 2024 to identify relevant studies published until the date of the search. We combined controlled vocabulary (MeSH Terms, Emtree terms) with keywords to produce a search engine for each database. Terms related to surgical flaps and genetic modifications delivered via bacterial, viral, or cellular vectors were implemented. The following Medical Subject Headings were used: genetic therapy, cell transplantation, mesenchymal stem cells, bacteria, and drug carriers. The full search strategy is available in the PROSPERO report.

### 4.2. Inclusion Criteria 

The following studies were included: studies that compared at least one control group with a treatment group, studies with different animal models, studies in which genetic modification of a tissue bulk of a living organism or genetic modification of cells/tissues later integrated with the former tissue bulk were used in the treatment group, and studies in which the control group received sham genetic modification or were treated using previously established standards.

### 4.3. Exclusion Criteria

Studies excluded were human studies, only in vitro studies that included split-thickness skin grafts, studies in which flaps received additional ischemia after the intervention, studies that lacked any intervention in the control group, studies reported in languages other than English, studies that did not report any unified measurement of flap survival, and studies with severe flaws in the reported data.

### 4.4. Primary Outcome

The primary outcome was the percentage of flap survival after *n* days post-intervention or the percentage of necrotic tissue within the flap. Data reported as a percentage of flap necrosis were later transformed into a percentage of flap survival to unify the overall outcome.

### 4.5. Secondary Outcomes

The secondary outcomes included the median number of vessels in the high-power field, mean number of vessels in the high-power field, number of capillaries per mm^2^, and improvement of other parameters treated with the modified flap. The results reported in values other than the median number of vessels in the high-power field were transformed (where applicable) into such data.

Titles and abstracts of studies retrieved using the search strategy were screened using the Rayyan QCRI to identify studies that met the inclusion criteria. All screenings were independently performed by two review team members (M.G., M.K.). Any disagreement regarding the eligibility of the studies was resolved through discussion with a third author when necessary (W.P.).

### 4.6. Data Extraction, Collection and Synthesis

All the data were extracted using a predefined extraction form. The following information was collected: authors, year, information about the animal model (species, strain, sex, weight), delivered drug (name, mode, and route of delivery, dose), and treatment (time of administration and assessment, flap type, and size). For continuous outcomes, the mean, standard deviation, and sample size for each study group were obtained. When necessary, data were extracted from the figures using WebPlotDigitizer v.4.6. If only standard errors were provided, they were transformed into standard deviations, multiplied by the square root of the sample size. To avoid unit of analysis issues, for studies with more than one intervention group and only one control group, we divided the number of animals in the control group by the number of intervention groups or combined data from the intervention groups if they were sufficiently similar. All changes are described in Appendix A.

### 4.7. Statistical Analysis

All analyses were performed using R version 4.3.1 and RStudio. Results for the primary outcome (flap survival) were presented as mean differences with 95% confidence intervals. Statistical heterogeneity was determined using the I2 statistic. For all outcomes, we used the DerSimonian–Laird random-effects model meta-analysis because we anticipated significant heterogeneity in the administration and assessment times as well as dosages, vectors, and flap types. As planned, we performed separate meta-analyses for each identified intervention and subgroup analysis for each flap and vector type. We performed a quantitative synthesis only if at least three comparisons reporting the outcomes of a given intervention were available. To ascertain the robustness of the findings, we performed three sensitivity analyses: (1) excluding studies assessed to be at a high risk of bias (i.e., scored less than 4 points on the SYRCLE checklist), (2) excluding data extracted from figures and not reported explicitly in tables or manuscripts, and (3) using an inverse-variance fixed-effect model rather than a random-effects model. A separate sensitivity analysis was planned based on the time between drug administration and flap elevation. Due to a variety of reported time gaps and a multitude of identified studies, we decided to perform a meta-regression for all studies evaluating the efficacy of VEGF. Other study characteristics were also included in this model, including administration time, assessment time, vector type, and route of administration. Three individual studies that reported rare routes of administration (intra-arterial, intrafascial, and topical) were excluded from the model to avoid generating spurious associations. Results are presented as effect estimates with 95% confidence intervals. For the subgroup analyses, we tested the interactions using the χ^2^ significance test. 

The secondary outcome (blood vessel density) was reported using different tools and measures in the included studies. Therefore, results in this case are presented as standardized mean differences (SMDs) using Hedge’s h with corresponding 95% CIs. We also performed a meta-regression to analyze how an overall effect estimate was influenced by potential moderators: administration time, assessment time, and intervention type. Results are presented as effect estimates with 95% confidence intervals. 

Publication bias was assessed for analyses including at least ten comparisons, funnel plots were generated and visually inspected, and a formal Egger’s test was performed. 

### 4.8. Risk of Bias (Quality) Assessment 

The quality of the included studies was determined using the SYRCLE checklist for study quality [65]. The two reviewers (M.G., W.P.) were blinded, and the quality of the studies was assessed separately. Discrepancies were discussed with a third reviewer to reach a consensus (A.S.).

### 4.9. Certainty of Evidence Assessment

The certainty of evidence was determined using the GRADE approach [66], modified for use in the preclinical meta-analysis [67]. M.G. and A.S. assessed the certainty of evidence independently. Any discrepancies and clinical relevance were consulted with the third reviewer (W.P.) to reach a consensus.

## 5. Conclusions

VEGF gene delivery to the flap increases its survival regardless of flap type, gene method, or timing in an in vivo model.Other proteins, such as FGF, HGF, and PDGF, may also be beneficial, but to a lesser extent.The quality of animal studies was predominantly low, which likely led to a high level of unexplained heterogeneity; however, general quantitative analysis is feasible.Translational studies appear to be sufficiently founded, yet any new interventions should be preceded by high-quality preclinical analysis.Additional, targeted, and detailed reviews and original studies should be performed to summarize the knowledge of pharmacodynamics and pharmacokinetics of gene therapies in flaps.

## Figures and Tables

**Figure 1 ijms-25-02622-f001:**
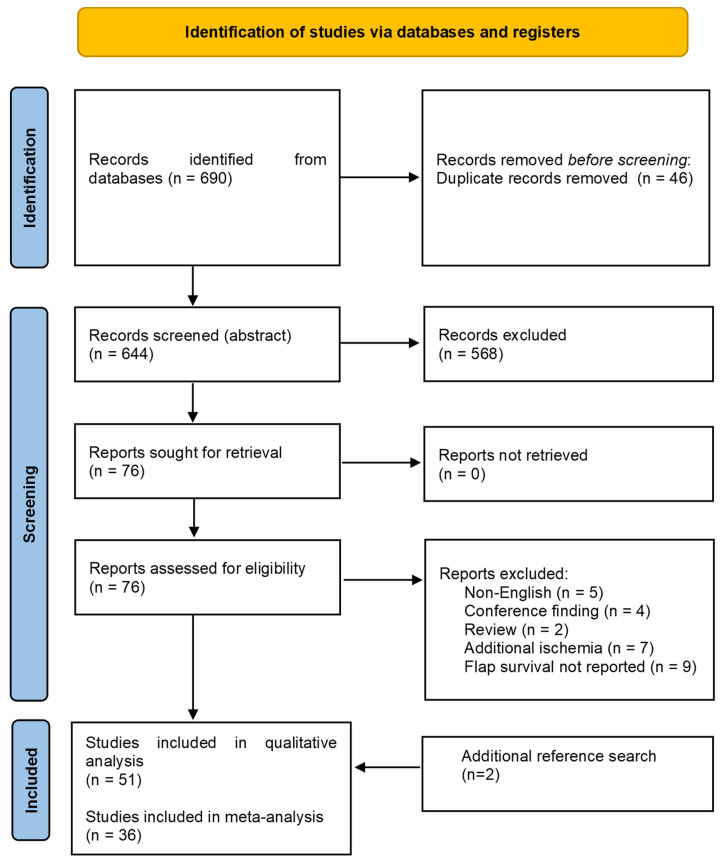
PRISMA Flowchart.

**Figure 2 ijms-25-02622-f002:**
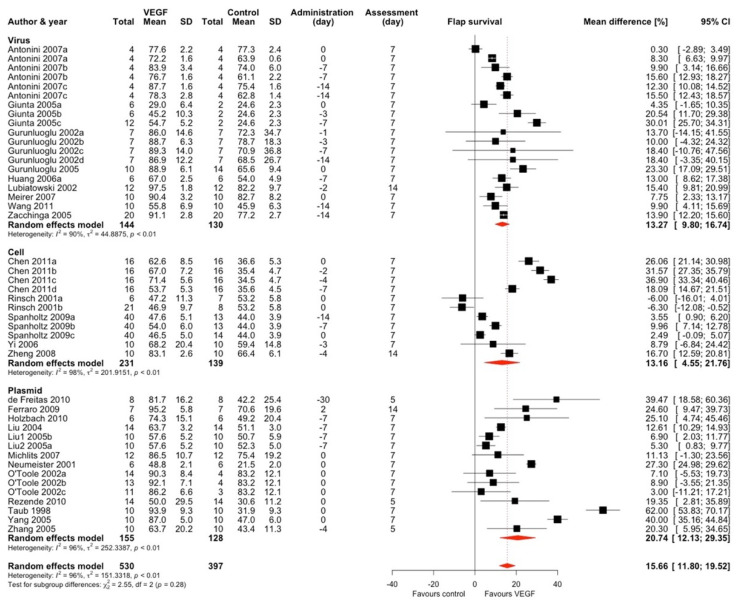
Mean difference in flap survival between VEGF treatment and control of included comparisons [7,8,9,10,11,12,13,15,16,28,29,30,32,33,34,37,38,39,40,41,42,43,45,46,47,48,50] with confidence intervals and effect estimates.

**Figure 3 ijms-25-02622-f003:**
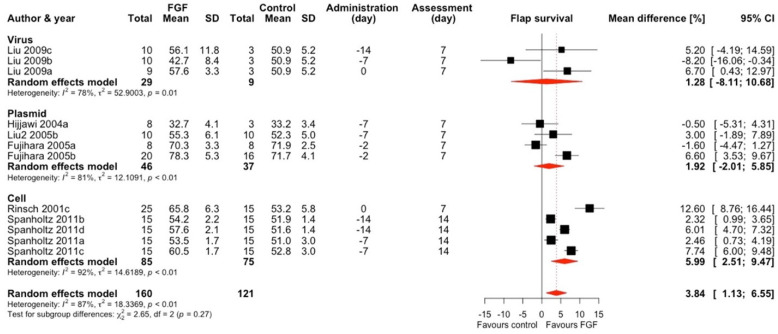
Mean difference in flap survival between FGF treatment and control of included comparisons [13,17,23,31,39,49] with confidence intervals and effect estimates.

**Figure 4 ijms-25-02622-f004:**
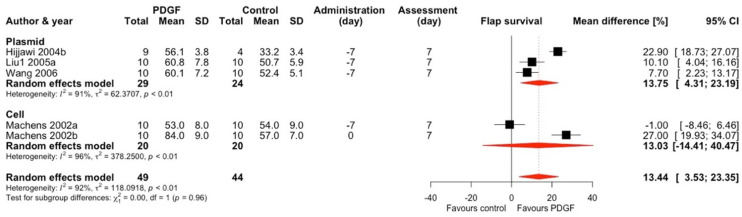
Mean difference in flap survival between PDGF treatment and control of included comparisons [11,17,18,44] with confidence intervals and effect estimates.

**Figure 5 ijms-25-02622-f005:**
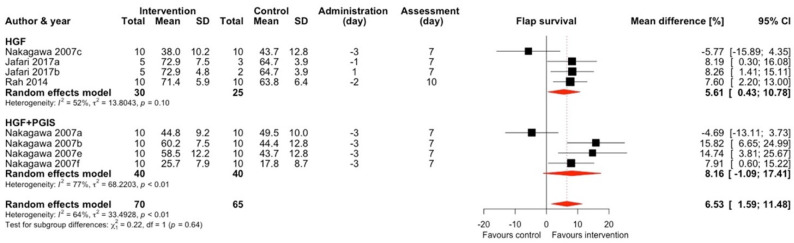
Mean difference in flap survival between HGF or HGF + PGIS treatment and control of included comparisons [19,20,53] with confidence intervals and effect estimates.

**Figure 6 ijms-25-02622-f006:**
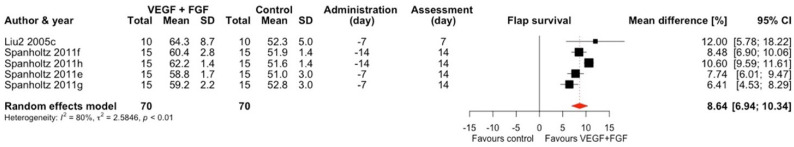
Mean difference in flap survival between VEGF + FGF treatment and control of included comparisons [13,49] with confidence intervals and effect estimates.

**Figure 7 ijms-25-02622-f007:**
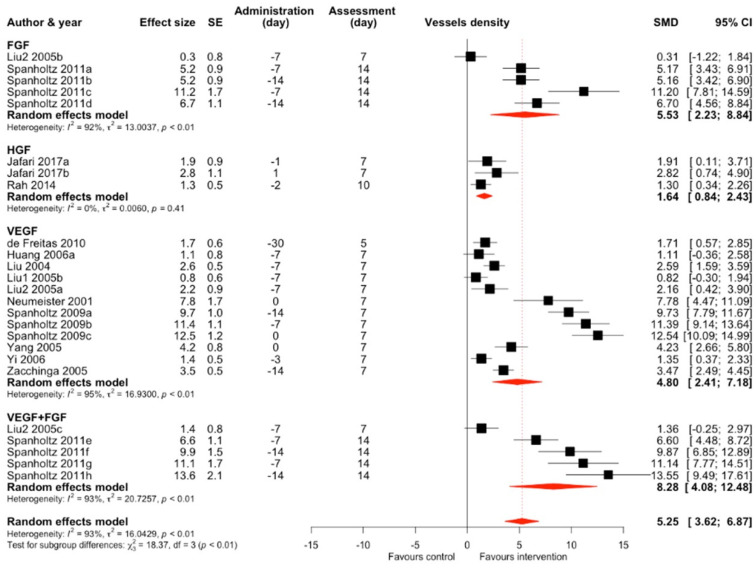
Standardized mean difference in vessel density between all interventions and control of included comparisons [7,11,12,13,15,20,33,41,42,45,47,49,53] with confidence intervals and effect estimates.

**Table 1 ijms-25-02622-t001:** Overview of studies included in primary outcome analysis. *—mode of expression based on gene expression assessment methods. **—if no standard deviation presented, results estimated based on graphical data available and not used for quantitative analysis.

First Author; Year (*)—Included In Meta-Analysis	Target	Vector	Mode of Expression *	Vector Quantity	Type of Control Group	Follow-Up	Flap Type	Animal Characteristics	Surgical Technique	Control Group Flap Survival (%, SD) **	Treatment Group Flap Survival (%, SD) **
Neumeister; 2001 (*) [7]	VEGF-165	Plasmid	Unspecified	50 μL	Lipofectamine only	7 days	Muscle	12 Wistar male rats (300–325 g)	Intramuscular injection after 4-h ischemia	21.5 ± 2.0	48.8 ± 2.1
Michlits; 2007 (*) [8]	VEGF-165	Plasmid	Unspecified	100 μg	No treatment (A) Fibrin-sealant (B) Fibrin-sealant and sham plasmid (C)	7 days	Epigastric	48 Sprague–Dawley male rats (300–350 g)	Topical administration after induced ischemia (not specified)	A—60.76 ± 10.20 B—74.30 ± 18.15 C—75.41 ± 19.19	86.54 ± 10.71
Meirer; 2007 (*) [9]	VEGF (I) Shockwave (II)	Virus	Unspecified	10^8^ PFU	No treatment	7 days	Epigastric	30 Sprague–Dawley male rats (300–500 g)	Subdermal injection	82.67 ± 8.15	I—90.42 ± 3.19 II—97.77 ± 4.34
Lubiatowski; 2002 (*) [10]	VEGF	Virus	Unspecified	10^8^ PFU	Sham virus (A) No treatment (B)	14 days	Epigastric	30 Sprague–Dawley male rats (300–350 g)	Subdermal injection two days before elevation	A—82.15 ± 9.7 B—79.9 ± 3.3	97.55 ± 1.84
Liu; 2005 (*) [11]	PDGF-B (I) VEGF (II)	Plasmid	Unspecified	50 μg	Saline (A) Sham plasmid (B)	7 days	McFarlane	45 Sprague–Dawley female rats (250–300 g)	Intradermal injection seven days before elevation	A—52.3 ± 5.0 B—50.7 ± 5.9	I—60.8 ± 7.8 II—57.6 ± 5.2
Liu; 2004 (*) [12]	VEGF-165	Plasmid	Unspecified	50 μg	Saline (A) Sham plasmid (B)	7 days	McFarlane	32 Sprague–Dawley female rats (250 g)	Intradermal injection seven days before elevation	A—51.3 ± 2.45 B—51.1 ± 3.05	63.71 ± 3.2
Liu; 2005 (*) [13]	VEGF-165 (I) FGF-β (II) PDGF-B (III)	Plasmid	Unspecified	50 μg	Sham plasmid	7 days	McFarlane	60 Sprague–Dawley female rats (250–300 g)	Intradermal injection 7 days before flap elevation	52.3 ± 5.0	I—57.6 ± 5.2 II—55.3 ± 6.1 I + II—64.3 ± 8.7 I + III—57.6 ± 9.5 I + II + III—44.9 ± 2.7
Lasso; 2007 [14]	VEGF-165	Cells (transduction)	Unspecified	5 × 10^5^ cells	Non-transduced cells	6 days	Axial ear	32 New Zealand white male rabbits (2500–3000 g)	Topical administration onto the scaffold with pedicle divided after five or two days (1, 2)	1—2.50 ± 7.07 2—51.25 ± 45.88	1—55.62 ± 38.95 2—95.62 ± 4.95
Huang; 2006 (*) [15]	VEGF-165 (I) eNOS (II)	Virus	Unspecified	5 × 10^8^ PFU	PBS (A) Sham virus (B)	7 days	McFarlane	24 Sprague–Dawley male rats (350–375 g)	Subdermal injection seven days before flap elevation	A—56.0 ± 3.0 B—54.0 ± 2.0	I—67.0 ± 1.0 II—70.0 ± 3.0
Holzbach; 2010 (*) [16]	VEGF-165	Plasmid (magnetized)	Unspecified	200 µl	Sham plasmid	7 days	McFarlane	46 Sprague–Dawley male rats (350 g)	Subcutaneous injection with additional ultrasounds seven days before flap elevation	49.2 ± 0.08	74.3 ± 0.05
Hijjawi; 2004 (*) [17]	PDGF-B (I) FGF-β (II)	Plasmid (collagen matrix)	Unspecified	4.8 mg	No treatment	7 days	Muscle	24 Sprague–Dawley male rats (250–300 g)	Subcutaneous injection seven days before flap elevation	33.2 ± 1.3	I—56.1 ± 1.25 II—32.7 ± 1.44
Machens; 2002 (*) [18]	PDGF-A	Cells (transduction)	Constant (up to 96 h)	1 × 10^7^ cells	Saline (A) Cell medium (B) Non-modified cells (C)	0 or 7 days (1,2)	Epigastric	80 Lewis inbred female rats (200–225 g)	Intramuscular at the time of flap elevation	A1—50.0 ± 9.0 A2—56.0± 8.0 B1—49.0 ± 11.0 B2—52.0 ± 10.0 C1—54.0 ± 9.0 C2—57.0 ± 7.0	1—53.0 ± 8.0 2—84.0 ± 9.0
Nakagawa; 2007 (*) [19]	HGF (I) PGIS (II)	Plasmid (jet injector)	Unspecified	400 µg	Sham plasmid	7 days	McFarlane	40 Sprague–Dawley male rats and 40 GK/Jcl male rats	Intracutaneous injection − 8 or 4 sites (1, 2) three days before flap elevation	1—43.73 ± 12.78 2—44.42 ± 10.02	I + II1—44.76 ± 9.17 I + II2—60.24 ± 7.45 I—37.96 ± 10.15 II—32.2 ± 11.64 I + II—58.47 ± 12.15
Rah; 2014 (*) [20]	HGF	Virus	Constant (up to 10 days)	1 × 10^7^ PFU	PBS (A) Sham protein (B)	10 days	McFarlane	30 Sprague–Dawley male rats (300–350 g)	Subdermal injection two days and immediately after flap elevation	A—39.2 ± 13.0 B—63.8 ± 6.4	71.4 ± 5.9
Huemer; 2004 [21]	TGF-β	Virus	Constant (up to 7 days)	1 × 10^8^ PFU	Saline (A) Sham virus (B)	7 days	Epigastric	30 Sprague–Dawley male rats (300–400 g)	Subdermal injection just prior to flap elevation	A—82.6 ± 4.6 B—82.2 ± 8.7	90.3 ± 4.0
Huemer; 2005 [22]	TGF-β (I) Shockwave (II)	Virus	Constant (up to 7 days)	1 × 10^8^ PFU	No treatment	7 days	Epigastric	30 Sprague–Dawley male rats (300–400 g)	Subdermal injection just prior to flap elevation	82.6 ± 4.3	I—90.3 ± 4.0 II—97.7 ± 1.8
Liu; 2009 (*) [23]	FGF-β	Virus	Unspecified	7.5 × 10^10^ PFU	Saline	7 days	McFarlane	38 Sprague–Dawley female rats (250–300 g)	Intradermal injection 14 days, 7 days, or at the time of flap elevation (1, 2, 3)	50.9 ± 5.2	1—56.1 ± 11.8 2—42.7 ± 8.4 3—57.6 ± 3.3
Lee; 2011 [24]	Relaxin	Virus	Constant (up to 10 days)	1 × 10^7^ PFU	PBS (A) Sham virus (B)	10 days	McFarlane	30 Sprague–Dawley male rats (300–350 g)	Subdermal injection two days before and immediately after flap elevation	A—37.0 ± 7.0 B—43.0 ± 3.0	55.0 ± 5.0
Jung; 2003 [25]	Ang1	Virus	Unspecified	1 × 10^8^ PFU	No treatment (A) Sham virus (B)	7 days	Epigastric	19 Sprague–Dawley male rats (200–250 g)	Subdermal injection two days before flap elevation	A—67.76 ± 5.98 B—78.65 ± 6.44	84.24 ± 6.54
Choi; 2020 [26]	DKK2	Virus	Constant (up to 10 days)	1 × 10^7^ PFU	No treatment (A) Sham virus (B)	10 days	McFarlane	30 Sprague–Dawley male rats (300–350 g)	Subdermal injection two days before and immediately after flap elevation	A—no results B—57.5 ± 4.21	80.0 ± 4.49
Lou; 2021 [27]	PLA2G4E	Virus	Constant (up to 7 days)	5 × 10^9^ PFU	Sham virus (measured here)	7 days	McFarlane	191 C57BL/6 J male mice (20–30g)	Subcutaneous injection of viral particles in 3 areas, 28 days prior to flap elevation	56.81 ± 8.03	81.57 ± 4.87
Gurunluoglu; 2002 (*) [28]	VEGF-164	Virus	Unspecified	10^8^ PFU	Saline (A) Sham virus (B)	7 days	Epigastric	84 Sprague–Dawley male rats (300–350 g)	Subdermal injections, 12 h, 3 days, 7 days, and 14 days prior to flap elevation (1, 2, 3, 4)	A1—68.3 ± 4.3 B1—72.3 ± 13.1 A2—65.7 ± 3.2 B2—78.7 ± 6.9 A3—63.8 ± 7.0 B3—70.9 ± 13.9 A4—69.6 ± 5.3 B4—68.5 ± 10.1	1—86.0 ± 5.5 2—88.7 ± 2.4 3—89.3 ± 5.3 4—86.9 ± 4.6
Gurunluoglu; 2005 (*) [29]	VEGF-121	Virus	Constant (up to 21 days)	10^8^ PFU	Saline (A) Sham virus (B)	7 days	Peninsular abdominal flap	34 Sprague–Dawley male rats (300–500 g)	Subdermal injection at the time of flap elevation	A—65.6 ± 9.4 B—56.0 ± 3.4	88.9 ± 6.1
Giunta; 2005 (*) [30]	VEGF-165	Virus	Constant (up to 7 days)	5 × 10^8^ PFU 1 × 10^9^ PFU(X)	Saline	7 days	Abdominal random-pattern	50 Sprague–Dawley male rats (350 g)	Subcutaneous injection at the time of flap elevation, 3 or 7 days before (1, 2, 3)	1—24.65 ± 2.27	1—29.0 ± 6.39 2—45.19 ± 10.32 3—48.5 ± 4.12 3 X—60.81 ± 6.25
Fujihara; 2005 (*) [31]	FGF-β	Plasmid	Constant (up to 3 days)	300 µg	Sham plasmid	7 days	Dorsal island skin	52 Sprague–Dawley male rats (200–250 g)	Intramuscular injection with additional electroporation (E) two days before flap elevation	71.9 ± 2.5 E—71.7 ± 4.1	70.3 ± 3.3 E—78.3 ± 5.3
Ferraro; 2009 (*) [32]	VEGF-165	Plasmid	Transient (at 12 days baseline expression)	100 µg	No treatment (A) Sham plasmid (B)	14 days	Single pedicle random	51 Sprague–Dawley male rats (275–300 g)	Intradermal injection (group with additional electroporation, E) two days before flap elevation	A—73.5 ± 4.1 BE—70.6 ± 7.4	65.6 ± 9.3 E—95.2 ± 2.2
De Freitas; 2010 (*) [33]	VEGF-165	Plasmid	Unspecified	100 µg	No treatment (A) PBS (B) Sham plasmid (C)	5 days	Muscle	32 Wistar male rats (350–400 g)	Intrafascial injection 30 days before flap elevation (during abdominoplasty) with electroporation in plasmid presence	A—75.35 ± 18.13 B—37.51 ± 28.06 C—42.2 ± 25.43	81.67 ± 16.20
Chen; 2011 (*) [34]	VEGF-165	Cells (transfection)	Transient (at 14 days baseline expression)	5 × 10^6^ cells	Non-transfected cells	7 days	Rectangular full-thickness	64 Wistar male and female rats (250–300 g)	Subdermal injection with internal control at different times before flap elevation (0, 2, 4, and 7 days—1, 2, 3, 4)	1—36.57 ± 5.28 2—35.43 ± 4.73 3—34.54 ± 4.68 4—35.62 ± 4.51	1—62.63 ± 8.53 2—67.00 ± 7.19 3—71.44 ± 5.56 4—53.71 ± 5.34
Chang; 2021 [35]	HIF1α	Plasmid	Transient (decrease after 48 h)	50 µg	Sham plasmid	7 days (1) 14 days (2)	McFarlane	20 Sprague–Dawley male rats	Intradermal injection seven days before flap elevation	1—68.75 ± 36.07 2—80.31 ± 21.25	1—92.97 ± 40.02 2—92.58 ± 15.07
Basu; 2014 [36]	VEGF-165	Plasmid	Unspecified	50 µg	No treatment	7 days (1) 14 days (2)	McFarlane	109 Sprague–Dawley male rats (275–300 g)	Intradermal injection at the time of flap elevation. Additional electrode array (E) during procedure.	1—84.75 ± 6.8 1E—84.75 ± 6.8 2E—79.8 ± 5.2	1—88.7 ± 4.84 1E—98.48 ± 1.87 2E—98.22 ± 2.82
Antonini; 2007 (*) [37]	VEGF-165	Virus	Unspecified	150 µL	Sham virus	14 days	Epigastric (Epi) Muscle (M)	48 Wistar male rats (260–290 g)	Subcutaneous/intramuscular injection at the time of flap elevation, 7 or 14 days prior (1, 2, 3)	Epi1—63.9 ± 0.6 Epi2—61.1 ± 2.2 Epi3—62.8 ± 1.4 M1—77.3 ± 1.2 M2—74 ± 3.0 M3—75.4 ± 0.8	Epi1—72.2 ± 1.6 Epi2—76.7 ± 1.5 Epi3—78.3 ± 2.8 M1—77.6 ± 1.1 M2—83.9 ± 1.7 M3—87.7 ± 0.8
Taub;1998 (*) [38]	VEGF-121	Plasmid	Unspecified	7 µg	Saline (A) Sham plasmid (B)	7 days	Epigastric	30 Sprague–Dawley female rats (250–300 g)	Intra-arterial injection during flap elevation	A—31.9 ± 12.64 B—28.1 ± 10.13	93.9 ± 4.91
Rinsch; 2001 (*) [39]	VEGF-121 (I) VEGF-165 (II) FGF-β (III)	Cells (transfection)	Constant	5 × 10^5^ cells	No treatment (A) Capsule only (B) Capsule containing non-modified cells (C)	7 days	McFarlane	86 Wistar female rats (250–300 g)	Administration of capsule on the subcutaneous tissue of flap during elevation	A—50.0 ± 3.8 B—44.9 ± 7.8 C—53.2 ± 5.8	I—47.2 ± 11.3 II—46.9 ± 9.7 III—65.8 ± 6.3
O’Toole; 2002 (*) [40]	VEGF-165 (I) VEGF-167 (II) VEGF-186 (III)	Plasmid	Constant (up to 2 days)	50 µg	Sham plasmid (A) Saline (B)	7 days	Epigastric	60 Sprague–Dawley male rats (400–550 g)	Subcutaneous injection at the time of flap elevation	A—83.2 ± 12.08 B—81.2 ± 12.08	I—90.3 ± 8.4 II—92.1 ± 7.09 III—86.2 ± 6.57
Yang; 2005 (*) [41]	VEGF-121	Plasmid	Unspecified	80 µg	Sham plasmid (A) Saline (B)	7 days	McFarlane	30 Sprague–Dawley female rats (280–320 g)	Intramuscular injection at the time of flap elevation	A—47.0 ± 6.0 B—46.0 ± 5.0	87.0 ± 5.0
Zacchigna; 2005 (*) [42]	VEGF-165	Virus	Unspecified	1.5 × 10^11^ PFU	Sham plasmid	7 days	Epigastric (Epi) Muscle (M)	88 Wistar male rats (250–300 g)	Subcutaneous (E) or intramuscular (M) injection at the time of flap elevation (1), 7 days before (2) or 14 days before (3)	1E—62.6 ± 1.8 2E—61.0 3E—63.0 1M—78.0 2M—75.0 3M—77.2 ± 2.7	1E—71.0 2E—77.0 3E—79.0 1M—78.0 2M—85.0 3M—91.1 ± 2.8
Zhang; 2005 (*) [43]	VEGF-165	Plasmid	Unspecified	50 µg	Sham plasmid (A) Saline (B)	4 days	Muscle	52 Sprague–Dawley rats (380–420 g)	Subcutaneous injection 4 days before flap elevation	A—36.3 ±13.1 B—43.4 ± 11.3	63.7 ± 20.2
Wang; 2006 (*) [44]	PDGF-B	Plasmid	Unspecified	50 µg	Saline	7 days	McFarlane	20 Sprague–Dawley female rats (250–300 g)	Intradermal injection at the time of flap elevation	52.4 ± 5.1	60.1 ± 7.2
Yi; 2006 (*) [45]	VEGF-165	Cells (transfection)	Transient (peak at day 4)	5 × 10^5^ cells	Non-transfected cells (A) culture medium (B)	28 days	Cranially based	30 athymic nude mice	Subcutaneous injection three days before flap elevation	A—59.37 ± 14.83 B—40.0 ± 16.65	68.16 ± 20.4
Zheng; 2008 (*) [46]	VEGF-165	Cells (transfection)	Transient (baseline expression at day 14)	5 × 10^6^ cells	Non-transfected cells (A) culture medium (B)	14 days	McFarlane	30 Sprague–Dawley rats (100–120 g)	Subcutaneous injection four days before flap elevation	A—66.4 ± 6.1 B—51.5 ± 7.5	83.1 ± 2.6
Spanholtz; 2009 (*) [47]	VEGF-165	Cells (transduction)	Transient (baseline expression at day 5)	1 × 10^7^ cells	Sham transduced cells (A) Non-transduced cells (B) saline (C)	7 days	McFarlane	80 Sprague–Dawley female rats (200–225 g)	Intradermal injection at different times before flap elevation: at the same time (1), 7 days (2), or 14 days before (3). Injection within the flap and into the surroundings	A—44.05 ± 3.92 B—45.0 C—47.0	1—47.60 ± 5.11 2—54.01 ± 5.95 3—46.54 ± 5.02
Rezende; 2010 (*) [48]	VEGF-165	Plasmid	Unspecified	50 µg	Empty plasmid in two different areas (A, B), sham plasmid (C) Saline (D)	5 days	Muscle	49 Wistar male rats (300 g)	Intradermal injection at the time of flap elevation with electroporation in different areas (* or **)	A—77.20 ± 9.79 B—61.38 ± 12.50 C—no data D—75.0 ± 1.0	*—60.77 ± 27.10 **—33.06 ± 31.76
Spanholtz; 2011 (*) [49]	FGF-β (I) VEGF-165 (II)	Cells (transfection)	Transient (baseline expression at day 14)	5 x10^6^ cells (*) 1 × 10^7^ cells(**)	Sham-modified fibroblasts (compared here) Non-modified fibroblast Cell medium only	14 days	McFarlane	320 Sprague–Dawley female rats (200–225 g)	Subdermal injections 7 days or 14 days (1, 2) before flap elevation within flap alone (*) and in flap surrounding and flap (**)	1 *—51.02 ± 2.96 2 *—51.90 ± 1.43 1 **—52.79 ± 3.01 2 **—51.56 ± 1.43	I + II * 1—58.76 ± 1.70 I + II * 2—60.38 ± 2.77 I + II ** 1—59.20 ± 2.17 I + II ** 2—62.16 ± 1.38 I * 1—53.48 ± 1.72 I * 2—54.22 ± 2.21 I ** 1—60.53 ± 1.68 I ** 2—57.57 ± 2.15
Wang; 2011 (*) [50]	VEGF-165	Virus	Unspecified	3 × 10^10^ PFU	Sham virus (A) Saline (B)	7 days	McFarlane	30 Sprague–Dawley female rats (250–300 g)	Intradermal injection 14 days before flap elevation	A—45.9 ± 6.3 B—48.7 ± 4.9	55.8 ± 6.9
Uemura; 2012 [51]	NF-kb	Oligonucleotide	Transient	1–2 mg	Sham oligonucleotide (A) No treatment (B)	5 days	Epigastric	36 Sprague–Dawley male rats (350–400 g)	Intra-arterial injection at the time of flap elevation	A—31.1 ± 3.7 B—31.7 ± 4.8	57.9 ± 8.4
Wang; 2013 [52]	KGF	Virus	Unspecified	1 × 10^9^ PFU	Sham virus with dexamethasone (A) Dexamethasone (B) PBS (C)	35 days	McFarlane	60 Sprague–Dawley male rats (300–350 g)	Subdermal injection in wound margin five days after flap elevation	A—86.19 ± 1.63 B—85.33 ± 1.98 C—84.20 ± 2.55	88.63 ± 0.69
Jafari; 2017 (*) [53]	HGF	Plasmid	Unspecified	25 µg	No treatment	7 days	McFarlane	15 Wistar male rats (290–320 g)	Intradermal injection 24 h before (1) or 24 h after (2) flap elevation followed by electroporation	64.67 ± 3.90	1—72.86 ± 7.46 2—2.93 ± 4.81
Jafari; 2018 [54]	IL-10	Plasmid	Unspecified	100 µg	No treatment	7 days	McFarlane	9 Wistar male rats (300–330 g)	Intradermal injection 24 h before flap elevation followed by electroporation	64.77 ± 3.90	81.26 ± 4.70
Jafari; 2021 [55]	IL-10 (I) HGF (II) VEGF-165 (III)	Plasmid	Unspecified	100 µg	No treatment	7 days	McFarlane	15 Wistar male rats (290–320 g)	Intradermal injection 24 h before flap (1) or 24 h (2) after flap elevation followed by electroporation	64.77 ± 3.90	I1 + III2—81.66 ± 9.70 I1 + II2—74.06 ± 1.65
Zhang; 2011 [56]	SDF-1	Cells (transduction) (I) Virus (II)	Unspecified	1 × 10^6^ cells (I) 6.25 × 10^9^ PFU (II)	Non-transduced MSC (A) saline (B)	10 days	Epigastric	24 Lewis rats (200–300 g)	Intravascular injection proximal to the anastomosis of femoral artery at the time of flap elevation	A—88.56 ± 3.72 B—79.12 ± 3.34	I—96.82 ± 2.35 II—86.86 ± 4.09
Luo; 2021 [57]	SDF-1α	Cells (modRNA transfection)	Constant (up to 14 days)	5.4 × 10^7^ cells	Sham transfected fibroblasts (A) PBS (B)	10 days (extended to 28 days for the treatment group)	McFarlane	60 Sprague–Dawley male rats (250–300 g)	Intradermal injection at the time of flap elevation	A—animals died B—animals died	after 28 days—92.2

Legend: VEGF—vascular endothelial growth factor, Shockwave—extracorporeal shockwave therapy, PDGF—platelet-derived growth factor, FGF-β—fibroblast growth factor 2, eNOS—endothelial nitric oxide synthase, HGF—hepatocyte growth factor, PGIS—prostacyclin synthase, TGF-β—transforming growth factor beta, Ang1—angiopoietin 1, DKK2—Dickkopf-related protein 2, PLA2G4E—phospholipase A2, HIF1α—hypoxia-inducible factor 1-alpha, NF-kb—nuclear factor kappa-light-chain-enhancer of activated B cells, KGF—keratinocyte growth factor, IL-10—interleukin 10, PFU—plaque-forming unit, PBS—phosphate-buffered saline, SDF-1—stromal cell-derived factor 1, MSC—mesenchymal stem cells, modRNA—modified RNA.

## Data Availability

Not applicable.

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
