# Peer review of "VEGF and Other Gene Therapies Improve Flap Survival—A Systematic Review and Meta-Analysis of Preclinical Studies"

_ijms, 2024, doi:10.3390/ijms25052622_

Round 1
Reviewer 1 Report
Comments and Suggestions for Authors
see file

minor editing
Reviewer 2 Report
Comments and Suggestions for Authors
This systematic review and meta-analysis is the first one to summarize the efficacy of surgical flap genetic modification based on the findings of animal studies. Although all the methodology followed was the proper one and four databases (PubMed, Embase, Web of Science, and 286 Scopus) were assessed, the last search was conducted on 11.12.22. This was a year ago, and the data should be more up-to-date. Otherwise, the presentation of the data, the subgroup analyses, and the results are clear and concise. Understanding how gene delivery to the flap contributes to its survival is crucial. However, studies of higher quality than those reported in this meta-analysis should be conducted in the future to draw safe results.
Author Response
Thank you for your time spent reviewing our article. We have updated our literature search. We have found two additional relevant articles, however both were only analyzed qualitatively. We updated all relevant figures (Figure 1 and Table 1) and parts of the manuscript (Abstract, Results) that contain that information. We also highlighted the conclusion to perform studies of higher quality in the Conclusion section.
Reviewer 3 Report
Comments and Suggestions for Authors
This meta-analysis of the role of gene therapy in the survival of flaps is well presented and conducted. Authors did a great job and used the correct analytical criteria to obtain conclusions. I recommend publication in the actual format, but the title of the work is limited to a generalist term while authors are mainly evaluating the role of VEGF.
Author Response
Thank you for your time and effort to review our manuscript. We used a general form of the title because we screened the literature not only for the VEGF treatment, but also other proteins. Indeed, most results focus on the VEGF as the most prevalently utilized approach. Yet, we altered the title to address your concern.